# ASSISTING THE ADVERSARY
# TO IMPROVE GAN TRAINING

## ABSTRACT

Some of the most popular methods for improving the stability and performance of GANs involve constraining or regularizing the discriminator. In this paper we consider a largely overlooked regularization technique which we refer to as the Adversary's Assistant (AdvAs). We motivate this using a different perspective to that of prior work. Specifically, we consider a common mismatch between theoretical analysis and practice: analysis often assumes that the discriminator reaches its optimum on each iteration. In practice, this is essentially never true, often leading to poor gradient estimates for the generator. To address this, AdvAs is a theoretically motivated penalty imposed on the generator based on the norm of the gradients used to train the discriminator. This encourages the generator to move towards points where the discriminator is optimal. We demonstrate the effect of applying AdvAs to several GAN objectives, datasets and network architectures. The results indicate a reduction in the mismatch between theory and practice and that AdvAs can lead to improvement of GAN training, as measured by FID scores.

## 1 INTRODUCTION

The generative adversarial network (GAN) framework (Goodfellow et al., 2014) trains a neural network known as a *generator* which maps from a random vector to an output such as an image. Key to training is another neural network, the *adversary* (sometimes called a discriminator or critic), which is trained to distinguish between "true" and generated data. This is done by *maximizing* one of the many different objectives proposed in the literature; see for instance Goodfellow et al. (2014); Arjovsky et al. (2017); Nowozin et al. (2016). The generator directly competes against the adversary: it is trained to *minimize* the same objective, which it does by making the generated data more similar to the true data. GANs are efficient to sample from, requiring a single pass through a deep network, and highly flexible, as they do not require an explicit likelihood. They are especially suited to producing photo-realistic images (Zhou et al., 2019) compared to competing methods like normalizing flows, which impose strict requirements on the neural network architecture (Kobyzev et al., 2020; Rezende & Mohamed, 2015) and VAEs (Kingma & Welling, 2014; Razavi et al., 2019; Vahdat & Kautz, 2020). Counterbalancing their appealing properties, GANs can have unstable training dynamics (Kurach et al., 2019; Goodfellow, 2017; Kodali et al., 2017; Mescheder et al., 2018).

Substantial research effort has been directed towards improving the training of GANs. These endeavors can generally be divided into two camps, albeit with significant overlap. The first develops better learning objectives for the generator/adversary to minimize/maximize. These are designed to have properties which improve training (Arjovsky et al., 2017; Li et al., 2017; Nowozin et al., 2016). The other camp develops techniques to regularize the adversary and improve its training dynamics (Kodali et al., 2017; Roth et al., 2017; Miyato et al., 2018). The adversary can then provide a better learning signal for the generator. Despite these contributions, stabilizing the training of GANs remains unsolved and continues to be an active research area.

An overlooked approach is to train the generator in a way that accounts for the adversary not being trained to convergence. One such approach was introduced by Mescheder et al. (2017) and later built on by Nagarajan & Kolter (2017). The proposed method is a regularization term based on the norm of the gradients used to train the adversary. This is motivated as a means to improve the convergence properties of the minimax game. The purpose of this paper is to provide a new perspective as to why

this regularizer is appropriate. Our perspective differs in that we view it as promoting updates that lead to a solution that satisfies a sufficient condition for the adversary to be optimal. To be precise, it encourages the generator to move towards points where the adversary's current parameters are optimal. Informally, this regularizer "assists" the adversary, and for this reason we refer to this regularization method as the *Adversary's Assistant* (AdvAs).

We additionally propose a version of AdvAs which is hyperparameter-free. Furthermore, we release a library which makes it simple to integrate into existing code. We demonstrate its application to a standard architecture with the WGAN-GP objective (Arjovsky et al., 2017; Gulrajani et al., 2017); the state-of-the-art StyleGAN2 architecture and objective introduced by Karras et al. (2020); and the AutoGAN architecture and objective introduced by Gong et al. (2019). We test these on the MNIST (Lecun et al., 1998), CelebA (Liu et al., 2015), CIFAR10 (Krizhevsky et al., 2009) datasets respectively. We show that AdvAs improves training on all datasets, as measured by the Fréchet Inception Distance (FID) (Heusel et al., 2017), and the inception score (Salimans et al., 2016) where applicable.

## 2 BACKGROUND

A generator is a neural network $g : \mathcal{Z} \to \mathcal{X} \subseteq \mathbb{R}^{d_x}$ which maps from a random vector $\boldsymbol{z} \in \mathcal{Z}$ to an output $\boldsymbol{x} \in \mathcal{X}$ (e.g., an image). Due to the distribution over $\boldsymbol{z}$, the function $g$ induces a distribution over its output $\boldsymbol{x} = g(\boldsymbol{z})$. If $g$ is invertible and differentiable, the probability density function (PDF) over $\boldsymbol{x}$ from this "change of variables" could be computed. This is not necessary for training GANs, meaning that no such restrictions need to be placed on the neural network $g$. We denote this distribution $p_\theta(\boldsymbol{x})$ where $\theta \in \Theta \subseteq \mathbb{R}^{d_g}$ denotes the generator's parameters. The GAN is trained on a dataset $\boldsymbol{x}_1, \ldots, \boldsymbol{x}_N$, where each $\boldsymbol{x}_i$ is in $\mathcal{X}$. We assume that this is sampled i.i.d. from a data-generating distribution $p_{\text{true}}$. Then the aim of training is to learn $\theta$ so that $p_\theta$ is as close as possible to $p_{\text{true}}$. Section 2.1 will make precise what is meant by "close."

The adversary $a_\phi : \mathcal{X} \to \mathcal{A}$ has parameters $\phi \in \Phi \subseteq \mathbb{R}^{d_a}$ which are typically trained alternately with the generator. It receives as input either the data or the generator's outputs. The set that it maps to, $\mathcal{A}$, is dependent on the GAN type. For example, Goodfellow et al. (2014) define an adversary which maps from $x \in \mathcal{X}$ to the probability that $x$ is a "real" data point from the dataset, as opposed to a "fake" from the generator. They therefore choose $\mathcal{A}$ to be $[0, 1]$ and train the adversary by maximizing the associated log-likelihood objective,

$$h(p_\theta, a_\phi) = \mathbb{E}_{x \sim p_{\text{true}}} \left[ \log a_\phi(x) \right] + \mathbb{E}_{x \sim p_\theta} \left[ \log(1 - a_\phi(x)) \right]. \tag{1}$$

Using the intuition that the generator should generate samples that seem real and therefore "fool" the adversary, the generator is trained to minimize $h(p_\theta, a_\phi)$. Since we find $\theta$ to minimize this objective while fitting $\phi$ to maximize it, training a GAN is equivalent to solving the minimax game,

$$\min_\theta \max_\phi h(p_\theta, a_\phi). \tag{2}$$

Eq. (1) gives the original form for $h(p_\theta, a_\phi)$ used by Goodfellow et al. (2014) but this form varies between different GANs, as we will discuss in Section 2.1. The minimization and maximization in Eq. (2) are performed with gradient descent in practice. To be precise, we define $L_{\text{gen}}(\theta, \phi) = h(p_\theta, a_\phi)$ and $L_{\text{adv}}(\theta, \phi) = -h(p_\theta, a_\phi)$. These are treated as losses for the generator and adversary respectively, and both are minimized. In other words, we turn the maximization of $h(p_\theta, a_\phi)$ w.r.t. $\phi$ into a minimization of $L_{\text{adv}}(\theta, \phi)$. Then on each iteration, $\theta$ and $\phi$ are updated one after the other using gradient descent steps along their respective gradients:

$$\boldsymbol{\nabla}_\theta L_{\text{gen}}(\theta, \phi) = \boldsymbol{\nabla}_\theta h(p_\theta, a_\phi), \tag{3}$$

$$\boldsymbol{\nabla}_\phi L_{\text{adv}}(\theta, \phi) = -\boldsymbol{\nabla}_\phi h(p_\theta, a_\phi). \tag{4}$$

### 2.1 GANS MINIMIZE DIVERGENCES

A common theme in the GAN literature is analysis based on what we call the *optimal adversary assumption*. This is the assumption that, before each generator update, we have found the adversary $a_\phi$ which maximizes $h(p_\theta, a_\phi)$ given the current value of $\theta$. To be precise, we define a class of permissible adversary functions $\mathcal{F}$. This is often simply the space of all functions mapping $\mathcal{X} \to \mathcal{A}$

(Goodfellow et al., 2014), but is in some GAN variants constrained by, e.g., a Lipschitz constant (Arjovsky et al., 2017). Then we call the adversary $a_\phi$ optimal for a particular value of $\theta$ if and only if $h(p_\theta, a_\phi) = \max_{a \in \mathcal{F}} h(p_\theta, a)$.

In practice, the neural network $a_\phi$ cannot represent every $a \in \mathcal{F}$ and so it may not be able to parameterize an optimal adversary for a given $\theta$. As is common in the literature, we assume that the neural network is expressive enough that this is not an issue, i.e., we assume that for any $\theta$, there exists at least one $\phi \in \Phi$ resulting in an optimal adversary. Then, noting that there may be multiple such $\phi \in \Phi$, we define $\Phi^*(\theta)$ to be the set of all optimal adversary parameters. That is, $\Phi^*(\theta) = \{\phi \in \Phi \mid h(p_\theta, a_\phi) = \max_{a \in \mathcal{F}} h(p_\theta, a)\}$ and the optimal adversary assumptions says that before each update of $\theta$ we have found $\phi \in \Phi^*(\theta)$. We emphasize that, in part due to the limited number of gradient updates performed on $\phi$, this assumption is essentially never true in practice. This paper presents a method to improve the training of GANs by addressing this issue.

The optimal adversary assumption simplifies analysis of GAN training considerably. Instead of being a two-player game, it turns into a case of minimizing an objective with respect to $\theta$ alone. We denote this objective

$$\mathcal{M}(p_\theta) = \max_{a \in \mathcal{F}} h(p_\theta, a) = h(p_\theta, a_{\phi^*}) \quad \text{where} \quad \phi^* \in \Phi^*(\theta). \tag{5}$$

For example, Goodfellow et al. (2014) showed that using the objective presented in Eq. (1) results in $\mathcal{M}(p_\theta) = 2 \cdot \text{JSD}(p_{\text{true}} || p_\theta) - \log 4$, where JSD is the Jensen-Shannon divergence. By making the optimal adversary assumption, they could prove that their GAN training procedure would converge, and would minimize the Jensen-Shannon divergence between $p_{\text{true}}$ and $p_\theta$.

A spate of research following the introduction of the original GAN objective has similarly made use of the optimal adversary assumption to propose GANs which minimize different divergences. For example, Wasserstein GANs (WGANs) (Arjovsky et al., 2017) minimize a Wasserstein distance. MMD GANs (Li et al., 2017) minimize a distance known as the maximum mean discrepancy. Nowozin et al. (2016) introduce f-GANs which minimize f-divergences, a class including the Kullback-Leibler and Jensen-Shannon divergences. We emphasize that this is by no means an exhaustive list. Like these studies, this paper is motivated by the perspective that, under the optimal adversary assumption, GANs minimize a divergence. However, the GAN framework can also be viewed from a more game-theoretic perspective (Kodali et al., 2017; Grnarova et al., 2018).

## 3 DOES AN OPTIMAL ADVERSARY LEAD TO OPTIMAL GRADIENTS?

As introduced above, the training of an adversary does not need to be considered in any analysis if it is simply assumed to always be optimal. From this perspective, the goal of training GANs can be seen as learning the generator to minimize $\mathcal{M}(p_\theta)$. This leads to the question: assuming that we have an optimal adversary, can we compute the gradient required for the generator update, $\nabla_\theta \mathcal{M}(p_\theta)$? To clarify, assume that we have generator parameters $\theta'$, and have found $\phi^* \in \Phi^*(\theta')$ such that $h(p_{\theta'}, a_{\phi^*})$ and $\mathcal{M}(p_{\theta'})$ are equal in value. We then want to take a gradient step on $\theta'$ to minimize $\mathcal{M}(p_{\theta'})$. Virtually all GAN methods do this by assuming that $\mathcal{M}(p_{\theta'})$ and $h(p_{\theta'}, a_{\phi^*})$ have equal gradients with respect to $\theta$ at $\theta'$. That is, it is assumed that $\nabla_\theta \mathcal{M}(p_\theta) \mid_{\theta=\theta'}$ is equal to the *partial derivative*[1] $D_1 h(p_\theta, a_{\phi^*}) \mid_{\theta=\theta'}$. It is not immediately obvious that this is true.

In the GAN literature this concern has largely been overlooked, with a few treatments for specific GAN types, see e.g. Arjovsky et al. (2017); Goodfellow et al. (2014). In particular, Arjovsky et al. (2017) invoke (but do not explicitly prove) an extension of Theorem 1 in Milgrom & Segal (2002) to prove that the Wasserstein GAN has optimal gradients if the adversary is optimal, i.e. $D_1 h(p_\theta, a_{\phi^*}) \mid_{\theta=\theta'} = \nabla_\theta \mathcal{M}(p_\theta) \mid_{\theta=\theta'}$. We note that this extension can, in fact, be used to prove that GANs in general have this property under fairly weak assumptions:

**Theorem 1.** *Let $\mathcal{M}(p_\theta) = h(p_\theta, a_{\phi^*})$ for any $\phi^* \in \Phi^*(\theta)$, as defined in Eq. (5). Assuming that $\mathcal{M}(p_\theta)$ is differentiable w.r.t. $\theta$ and $h(p_\theta, a_\phi)$ is differentiable w.r.t. $\theta$ for all $\phi \in \Phi^*(\theta)$, then if $\phi^* \in \Phi^*(\theta)$ we have*

$$\nabla_\theta \mathcal{M}(p_\theta) = D_1 h(p_\theta, a_{\phi^*}). \tag{6}$$

---

[1]We use $D_1 h(p_\theta, a_\phi)$ to denote the partial derivative of $h(p_\theta, a_\phi)$ with respect to $\theta$ with $\phi$ kept constant. Similarly, we will use $D_2 h(p_\theta, a_\phi)$ to denote the derivative of $h(p_\theta, a_\phi)$ with respect to $\phi$, with $\theta$ held constant.

See Appendix D.1 for a proof. We emphasize Theorem 1 applies only if the adversary is optimal. If this is not the case we cannot quantify, and so cannot directly minimize or account for, the discrepancy between $\boldsymbol{\nabla}_\theta \mathcal{M}(p_\theta)$ and $D_1 h(p_\theta, a_{\phi^*})$. Instead of attempting to do so, we consider an approach that drives the parameters towards regions where $\phi^* \in \Phi^*(\theta)$ so that Theorem 1 can be invoked.

## 3.1 Adversary constructors

To see how we may impose the constraint that Eq. (6) is true, we consider a trivial relationship between any generator and the corresponding optimal adversary. If an optimal adversary exists for every $\theta \in \Theta$ then there exists some, possibly non-unique, function $f : \Theta \to \Phi$ that maps from any generator to a corresponding optimal adversary. That is, for all $\theta \in \Theta$, $f(\theta) = \phi^* \in \Phi^*(\theta)$ in which case $h(p_\theta, a_{f(\theta)}) = \max_{a \in \mathcal{F}} h(p_\theta, a)$. We refer to such a function as an *adversary constructor*. In an ideal scenario, we could compute the output of an adversary constructor, $f(\theta)$, for any $\theta$. We could then invoke Theorem 1 and the generator could be updated with the gradient $\boldsymbol{\nabla}_\theta \mathcal{M}(p_\theta) = D_1 h(p_\theta, a_{f(\theta)})$. In practice, computing $f(\theta)$ is infeasible and we can only approximate the optimal adversary parameters with gradient descent. There is therefore a mismatch between GAN theory, where Theorem 1 is often invoked, and practice, where the conditions to invoke it are essentially never satisfied. How then, can we address this problem? We look to the adversary constructors, which provide a condition that must be satisfied for the optimal adversary assumption to be true. Adversary constructors allow us to account for the influence of $\theta$ on $\phi$ by considering the total derivative $\boldsymbol{\nabla}_\theta h(p_\theta, a_{f(\theta)})$. We prove in Appendix D.2 that a comparison with the result of Theorem 1 leads to Corollary 1. In the next section, we motivate AdvAs as an attempt to fulfill a condition suggested by this corollary.

**Corollary 1.** *Let $f : \Theta \to \Phi$ be a differentiable mapping such that for all $\theta \in \Theta$, $\mathcal{M}(p_\theta) = h(p_\theta, a_{f(\theta)})$. If the conditions in Theorem 1 are satisfied and the Jacobian matrix of $f$ with respect to $\theta$, $\boldsymbol{J}_\theta(f)$ exists for all $\theta \in \Theta$ then*

$$D_2 h(p_\theta, a_{f(\theta)})^\mathrm{T} \boldsymbol{J}_\theta(f) = 0. \tag{7}$$

## 4 Assisting the adversary

Corollary 1 tells us that $D_2 h(p_\theta, a_\phi)^\mathrm{T} \boldsymbol{J}_\theta(f)$ will be zero whenever Theorem 1 can be invoked. This makes Eq. (7) a necessary, but not sufficient, condition for the invocation of Theorem 1. This suggests that the magnitude of $D_2 h(p_\theta, a_\phi)^\mathrm{T} \boldsymbol{J}_\theta(f)$ could be a measure of how "close" $D_1 h(p_\theta, a_{\phi^*})$ is to the desired gradient $\boldsymbol{\nabla}_\theta \mathcal{M}(p_\theta)$. However, the Jacobian $\boldsymbol{J}_\theta(f)$ is not tractable so $D_2 h(p_\theta, a_\phi)^\mathrm{T} \boldsymbol{J}_\theta(f)$ cannot be computed. The only term we can calculate in practice is $D_2 h(p_\theta, a_\phi)$, exactly the gradient used to train the adversary. If $D_2 h(p_\theta, a_\phi)$ is zero, then $D_2 h(p_\theta, a_\phi)^\mathrm{T} \boldsymbol{J}_\theta(f)$ is zero. The magnitude of $D_2 h(p_\theta, a_\phi)$ could therefore be an approximate measure of "closeness" instead of $D_2 h(p_\theta, a_\phi)^\mathrm{T} \boldsymbol{J}_\theta(f)$. This leads to an augmented generator loss, which regularizes generator updates to reduce the magnitude of $D_2 h(p_\theta, a_\phi)$. It has a scalar hyperparameter $\lambda \geq 0$, but Section 4.3 provides a heuristic which can remove the need to set this hyperparameter.

$$L_\mathrm{gen}^\mathrm{AdvAs}(\theta, \phi) = L_\mathrm{gen}(\theta, \phi) + \lambda r(\theta, \phi), \tag{8}$$

with

$$r(\theta, \phi) = \|\boldsymbol{\nabla}_\phi L_\mathrm{adv}(\theta, \phi)\|_2^2, \tag{9}$$

recalling that $\boldsymbol{\nabla}_\phi L_\mathrm{adv}(\theta, \phi) = -D_2 h(p_\theta, a_\phi)$. We emphasize that $r(\theta, \phi)$ is the same as that found in previous work (Mescheder et al., 2017; Nagarajan & Kolter, 2017).

Figuratively, AdvAs changes the generator updates to move in a conservative direction that does not over-exploit the adversary's sub-optimality. Consider the generator and adversary as two players attempting to out-maneuver one another. From Eq. (2), we see that the generator should learn to counteract the best possible adversary, rather than the current adversary. If the current adversary is sub-optimal, allowing it to catch up would yield better updates to the generator. One way to achieve this is to update the generator in a way that helps make the current adversary optimal. This behavior is exactly what AdvAs encourages. In this sense, it assists the adversary, leading to it's name, the Adversary's Assistant. We emphasize that using AdvAs involves making only a small modification to a GAN training algorithm, but for completeness we include pseudocode in Appendix A.

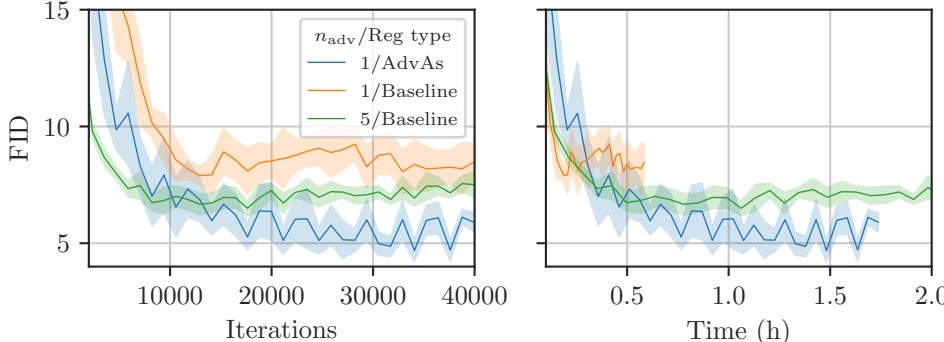

Figure 1: FID scores throughout training for the WGAN-GP objective on MNIST, estimated using 60 000 samples from the generator. We plot up to a maximum of 40 000 iterations. When plotting against time (right), this means some lines end before the two hours we show. The blue line shows the results with AdvAs, while the others are baselines with different values of $n_{\text{adv}}$.

### 4.1 ADVAS PRESERVES CONVERGENCE RESULTS

AdvAs has several desirable properties which support its use as a regularizer: (1) it does not interfere with the update on $\phi$, and recall that perfectly optimizing $\phi$ leads to $h(p_\theta, a_\phi) = \mathcal{M}(p_\theta)$. (2) Under mild conditions, $\boldsymbol{\nabla}_\phi r(\theta; \phi) \mid_{\phi=\phi^*}$ is zero for an optimal $\phi^*$ and so $\boldsymbol{\nabla}_\theta L_{\text{gen}}^{\text{AdvAs}}(\theta, \phi^*) = \boldsymbol{\nabla}_\theta \mathcal{M}(p_\theta)$. These properties imply that, under the optimal adversary assumption, optimizing $L_{\text{gen}}^{\text{AdvAs}}$ is in fact equivalent to optimizing $L_{\text{gen}}$. See Appendix D.4 for a proof. Therefore any convergence analysis which relies on the optimal adversary assumption is equally applicable when AdvAs is included in the loss. Regarding the mild conditions in property (2), we require that $\phi^*$ be a stationary point of $h(p_\theta, a_\phi)$. This is true as long as $h$ is differentiable w.r.t. $\phi$ at $(\theta, \phi^*)$ and $\phi^*$ does not lie on a boundary of $\Phi$. The optimal adversary parameters, $\phi^*$, cannot lie on a boundary unless, for example, weight clipping is used as in Arjovsky et al. (2017). In such cases, we cannot speak to the efficacy of applying AdvAs.

We make the additional observation that for some GAN objectives, minimizing $r(\theta, \phi)$ alone (as opposed to $L_{\text{gen}}$ or $L_{\text{gen}}^{\text{AdvAs}}$) may match $p_\theta$ and $p_{\text{true}}$. We show this in Appendix D.3 for the WGAN objective (Arjovsky et al., 2017). In particular, for all $\phi \in \Phi$, $r(\theta, \phi)$ is zero and at a global minimum whenever $p_\theta = p_{\text{true}}$. Experimental results in Appendix C.3 support this observation. However, the results appear worse than those obtained by optimizing either $L_{\text{gen}}$ or $L_{\text{gen}}^{\text{AdvAs}}$.

### 4.2 ESTIMATING THE ADVAS LOSS

It is not always possible, and seldom computationally feasible, to compute the AdvAs regularization term $r(\theta, \phi)$ exactly. We instead use a stochastic estimate. This is computed by simply estimating the gradient $\boldsymbol{\nabla}_\phi L_{\text{adv}}(\theta, \phi)$ with a minibatch and then taking the squared L2-norm of this gradient estimate. That is, defining $\tilde{L}_{\text{adv}}(\theta, \phi)$ as an unbiased estimate of the adversary's loss, we estimate $r(\theta, \phi)$ with

$$\tilde{r}(\theta, \phi) = \left\| \boldsymbol{\nabla}_\phi \tilde{L}_{\text{adv}}(\theta, \phi) \right\|_2^2. \tag{10}$$

Although the gradient estimate is unbiased, taking the norm results in a biased estimate of $r(\theta, \phi)$. However, comparisons with a more computationally expensive unbiased estimate[2] did not reveal a significant difference in performance.

---

[2]Computing an unbiased estimate can be done using the following: consider two independent and unbiased estimates of $\boldsymbol{\nabla}_\phi L_{\text{adv}}(\theta, \phi)$ denoted $\boldsymbol{X}, \boldsymbol{X}'$. Then $\mathbb{E}\left[\boldsymbol{X}^{\text{T}}\boldsymbol{X}'\right] = \mathbb{E}\left[\boldsymbol{X}\right]^{\text{T}} \mathbb{E}\left[\boldsymbol{X}'\right] = \left\|\boldsymbol{\nabla}_\phi L_{\text{adv}}(\theta, \phi)\right\|_2^2$. This implies that multiplying two estimates using independent samples is unbiased.

## 4.3 Removing the hyperparameter $\lambda$

Eq. (8) introduces a hyperparameter, $\lambda$, which we would prefer not to perform a grid-search on. Setting $\lambda$ to be too great can destabilize training. Conversely, setting it to be too small gives similar results to not using AdvAs. We therefore introduce a heuristic which can be used to avoid setting the hyperparameter. Our experiments suggests that this is often a good choice, although manually tuning $\lambda$ may yield greater gains. This heuristic involves considering the magnitudes of three gradients, and so we first define the notation,

$$\boldsymbol{g}_{\text{orig}}(\theta, \phi) = \boldsymbol{\nabla}_\theta L_{\text{gen}}(\theta, \phi),$$
$$\boldsymbol{g}_{\text{AdvAs}}(\theta, \phi) = \boldsymbol{\nabla}_\theta \tilde{r}(\theta, \phi),$$
$$\boldsymbol{g}_{\text{total}}(\theta, \phi, \lambda) = \boldsymbol{\nabla}_\theta L_{\text{gen}}^{\text{AdvAs}}(\theta, \phi)$$
$$= \boldsymbol{g}_{\text{orig}}(\theta, \phi) + \lambda \boldsymbol{g}_{\text{AdvAs}}(\theta, \phi).$$

The heuristic can be interpreted as choosing $\lambda$ at each iteration to prevent the total gradient, $\boldsymbol{g}_{\text{total}}(\theta, \phi, \lambda)$, from being dominated by the AdvAs term. Specifically, we ensure the magnitude of $\lambda \boldsymbol{g}_{\text{AdvAs}}(\theta, \phi)$ is less than or equal to the magnitude of $\boldsymbol{g}_{\text{orig}}$ by setting

$$\lambda = \min\left(1, \frac{\|\boldsymbol{g}_{\text{orig}}(\theta, \phi)\|_2}{\|\boldsymbol{g}_{\text{AdvAs}}(\theta, \phi)\|_2}\right) \qquad (11)$$

at every iteration. We then perform gradient descent along $\boldsymbol{g}_{\text{total}}(\theta, \phi, \lambda)$. This technique ensures that $\lambda$ is bounded above by 1.

Table 1: FID and IS scores on CIFAR10 using AutoGAN with and without AdvAs.

|  | IS $\pm\sigma$ | FID $\pm\sigma$ |
|---|---|---|
| AdvAs ($n_{\text{adv}} = 2$) | $8.4 \pm 0.1$ | $14.5 \pm 1.0$ |
| Baseline ($n_{\text{adv}} = 5$) | $8.3 \pm 0.1$ | $15.0 \pm 0.7$ |

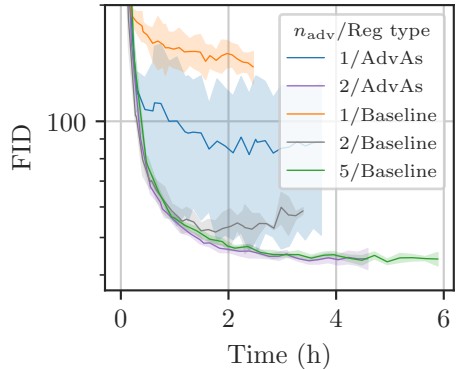

Figure 2: FID scores on CIFAR10 using Auto-GAN from baselines and AdvAs plotted with a log y-axis against running time for different values of $n_{\text{adv}}$. We see that AdvAs with $n_{\text{adv}} = 2$ yields the lowest FID scores at every point during training.

## 5 Experiments

We demonstrate the effect of incorporating AdvAs into GAN training using several GAN architectures, objectives, and datasets. Our experiments complement those of Nagarajan & Kolter (2017) and Mescheder et al. (2017). In each case, we compare GANs trained with AdvAs with baselines that do not use AdvAs but are otherwise identical. We first demonstrate the use of AdvAs in conjunction with the WGAN-GP objective (Gulrajani et al., 2017) to model MNIST (Lecun et al., 1998). In this experiment, we compare the performance gains achieved by AdvAs to a reasonable upper bound on the gains achievable with this type of regularization. We further support these findings with experiments on CIFAR10 (Krizhevsky et al., 2009) using AutoGAN (Gong et al., 2019), an architecture found through neural architecture search. We then demonstrate that AdvAs can improve training on larger images using StyleGAN2 (Karras et al., 2020) on CelebA (Liu et al., 2015). We quantify each network's progress throughout training using the FID score (Heusel et al., 2017). Since AdvAs increases the computation time per iteration, we plot training progress against time for each experiment. We also present inception scores (IS) (Salimans et al., 2016) where applicable. We estimate scores in each case with 5 random seeds and report the standard deviation ($\sigma$) as a measure of uncertainty.

AdvAs aims to improve performance by coming closer to having an optimal adversary. Another common way to achieve this is to use a larger number of adversary updates ($n_{\text{adv}}$) before each generator update. For each experiment, we show baselines with the value of $n_{\text{adv}}$ suggested in the literature. Noting that the computational complexity is $\mathcal{O}(n_{\text{adv}})$ and so keeping $n_{\text{adv}}$ low is desirable, we find that AdvAs can work well with lower values of $n_{\text{adv}}$ than the baseline. For a fair comparison, we also report baselines trained with these values of $n_{\text{adv}}$.

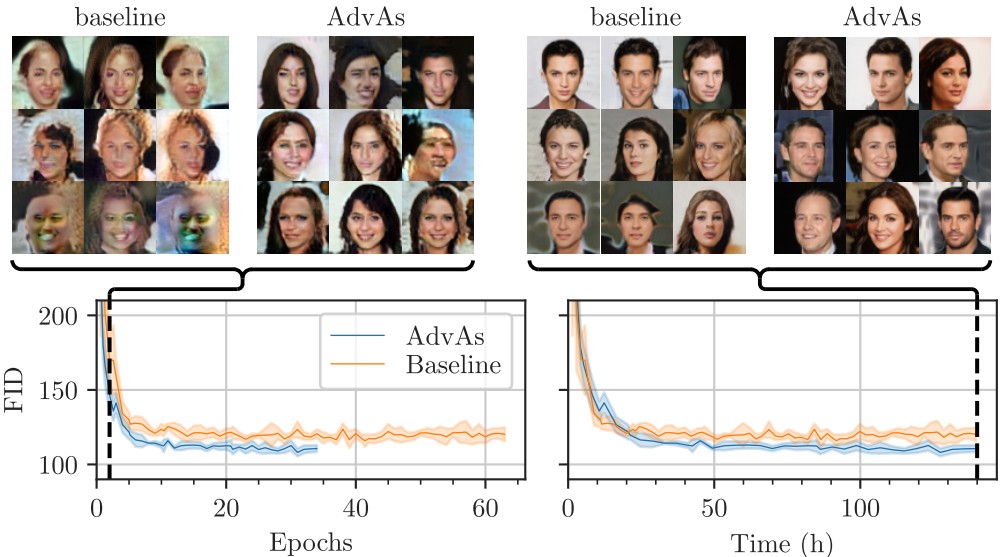

Figure 3: **Bottom:** FID scores throughout training estimated with 1000 samples, plotted against number of epochs (left) and training time (right). FID scores for AdvAs decrease more on each iteration at the start of training and converge to be 7.5% lower. **Top:** The left two columns show uncurated samples with and without AdvAs after 2 epochs. The rightmost two columns show uncurated samples from networks at the end of training. In each grid of images, each row is generated by a network with a different training seed and shows 3 images generated by passing a different random vector through this network. AdvAs leads to obvious qualitative improvement early in training.

For MNIST and CelebA, we avoid setting the hyperparameter $\lambda$ by using the heuristic proposed in Section 4.3. We found for CIFAR10 that manually tuning $\lambda$ gave better performance, and so set $\lambda = 0.01$. Additionally, on MNIST and CelebA, the methods we consider use regularization in the form of a gradient penalty (Gulrajani et al., 2017) for training the adversary. This is equivalent to including a regularization term $\gamma_{\mathrm{adv}}(\phi)$ in the definition of $L_{\mathrm{adv}}$. That is, $L_{\mathrm{adv}}(\theta, \phi) = -\boldsymbol{\nabla}_\phi h(p_\theta, a_\phi) + \gamma_{\mathrm{adv}}(\phi)$. Following Eq. (9) this regularization term is included in the AdvAs term $r(\theta, \phi)$. Another practical detail is that AutoGAN and StyleGAN2 are trained with a hinge loss (Lim & Ye, 2017). That is, when computing the adversary's loss $L_{\mathrm{adv}}(\theta, \phi)$, its output $a_\phi(\boldsymbol{x})$ is truncated to be below $+1$ for real images, or above $-1$ for generated images. This prevents it from receiving gradient feedback when its predictions are both accurate and confident. Intuitively, this stops its outputs becoming too large and damaging the generator's training. However, this truncation is not present when updating the generator. This means that the generator minimizes a different objective to the one maximized by the adversary, and so it is not exactly a minimax game. It is not clear that it is beneficial to calculate the AdvAs regularization term using this truncation. We found that better performance was obtained by computing $r(\theta, \phi)$ without truncation, and do this in the reported experiments.

## 5.1 WGAN-GP ON MNIST

We use a simple neural architecture: the generator consists of a fully-connected layer followed by two transposed convolutions. The adversary has three convolutional layers. Both use instance normalization (Ulyanov et al., 2017) and ReLU non-linearities; see Appendix E for details. We compare using AdvAs with $n_{\mathrm{adv}} = 1$ against the baseline for $n_{\mathrm{adv}} \in \{1, 5\}$ where $n_{\mathrm{adv}} = 5$ is suggested by Gulrajani et al. (2017). Fig. 1 shows the FID scores for each method throughout training. We see that using AdvAs with $n_{\mathrm{adv}} = 1$ leads to better performance on convergence; even compared to the baseline with $n_{\mathrm{adv}} = 5$, the best FID score reached is improved by 28%.

## 5.2 AUTOGAN ON CIFAR10

We next experiment on the generation of CIFAR10 (Krizhevsky et al., 2009) images. We use Auto-GAN (Gong et al., 2019), which has a generator architecture optimized for CIFAR10 using neural architecture search. It is trained with a hinge loss, as described previously, an exponential moving

average of generator weights, and typically uses $n_{\text{adv}} = 5$. Figure 2 shows FID scores throughout training for various values of $n_{\text{adv}}$, with and without AdvAs, each computed with 1000 samples. Table 1 shows FID scores at the end of training for the best performing value of $n_{\text{adv}}$ for each method, estimated with $50\,000$ samples. For a fixed $n_{\text{adv}}$ of either 1 or 2, using AdvAs improves the FID score. In fact, with $n_{\text{adv}} = 2$, the performance with AdvAs is indistinguishable from the baseline with the suggested setting of $n_{\text{adv}} = 5$. Unlike for MNIST, AdvAs does not outperform the baseline with high enough $n_{\text{adv}}$. We hypothesize that this is because, with an architecture highly optimized for $n_{\text{adv}} = 5$, the adversary is closer to being optimal when trained with $n_{\text{adv}} = 5$. Assuming this is the case, we would not expect AdvAs to improve training compared to a baseline with sufficient $n_{\text{adv}}$. Still, our results show that applying AdvAs allows the same performance with a lower $n_{\text{adv}}$.

### 5.3 STYLEGAN2 ON CELEBA

To demonstrate that AdvAs improves state-of-the-art GAN architectures and training procedures, we consider StyleGAN2 (Karras et al., 2020). We train this as proposed by Karras et al. (2020) with a WGAN-like objective with gradient penalty (Gulrajani et al., 2017), an exponential moving average of the generator weights, and various forms of regularization including path length, R1, and style-mixing regularization. More detail on these can be found in Karras et al. (2020), but we merely wish to emphasize that considerable effort has been put into tuning this training procedure. For this reason, we do not attempt to further tune $n_{\text{adv}}$, which is 1 by default. Any improvements from applying AdvAs indicate a beneficial effect not provided by other forms of regularization used.

Figure 3 compares the training of StyleGAN2 on CelebA at $64\times64$ resolution with and without the AdvAs regularizer. Using AdvAs has two main effects: (1) the generated images show bigger improvements per epoch at the start of training; and (2) the final FID score is improved by $7.5\%$. Even accounting for its greater time per iteration, the FID scores achieved by AdvAs overtake the baseline after one day of training. We verify that the baseline performance is similar to that reported by Zhou et al. (2019) with a similar architecture.

## 6 RELATED WORK

We motivated AdvAs from the perspective of the optimal adversary assumption. In this sense, it is similar to a large body of work aiming to improve and stabilize GAN training by better training the adversary. AdvAs differs fundamentally due to its focus on the training the generator rather than the adversary. This other work generally affects the discriminator in one of two broad ways: weight constraints and gradient penalties Brock et al. (2019). Weight normalization involves directly manipulating the parameters of the adversary, such as through weight clipping (Arjovsky et al., 2017) or spectral normalization (Miyato et al., 2018). Gradient penalties (Kodali et al., 2017; Roth et al., 2017; Gulrajani et al., 2017) impose soft constraints on the gradients of the adversary's output with respect to its input. Various forms exist with different motivations; see Mescheder et al. (2018) for a summary and analysis. AdvAs may appear similar to a gradient penalty, as it operates on gradients of the adversary. However, the gradients are w.r.t. the adversary's parameters rather than its input. Furthermore, AdvAs is added to the generator's loss and not the adversary's.

Regularizing generator updates has recently received more attention in the literature (Chu et al., 2020; Zhang et al., 2019; Brock et al., 2019). Chu et al. (2020) show theoretically that the effectiveness of different forms of regularization for both the generator and adversary is linked to the smoothness of the objective function. They present a set of conditions on the generator and adversary that ensure a smooth objective function, which they argue will stabilize GAN training. However, they leave the imposition of the required regularization on the generator to future work. Zhang et al. (2019) and Brock et al. (2019) consider applying spectral normalization (Miyato et al., 2018) to the generator, and find empirically that this improves performance.

## 7 DISCUSSION AND CONCLUSIONS

We have shown that AdvAs addresses the mismatch between theory, where the adversary is assumed to be trained to optimality, and practice, where this is never the case. We show improved training across three datasets, architectures, and GAN objectives, indicating that it successfully reduces this

disparity. This can lead to substantial improvements in final performance. We note that, while applying AdvAs in preliminary experiments with BEGAN (Berthelot et al., 2017) and LSGAN (Mao et al., 2017), we did not observe either a significant positive effect, or a significant negative effect other than the increased time per iteration. Nevertheless, AdvAs is simple to apply and will, in many cases, improve both training speed and final performance.

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
