# OpenReview forum: "Assisting the Adversary to Improve GAN Training"
_ICLR.cc/2021/Conference — Reject_

### Official Review · AnonReviewer3 · 2020-10-28
**A nice investigation, but needed to be more rigorous**

**Rating:** 4
**Confidence:** 4

**Review:**

This paper concerns how to efficiently regularize the generatior for training generative adversarial networks (GANs). A new regularizer for the generator loss is proposed to penalize the norm of the gradient with respect to discriminator’s parameters ($\phi$). In other words, the generator learns to encourage small norm of the discriminator’s gradient w.r.t $\phi$. The author(s) also propose a heuristic to remove the introduction of a further hyperparameter. The author(s) applied the proposed regularizer to WGAN-GP, AutoGAN, and StyleGAN2 to validate its effectiveness. Their experiments reveal that the new regularizer is promising.

Pros:
- The idea of penalizing the gradient w.r.t discriminator’s parameters when learning the generator is interesting. It could be great if the authors could explain more about its implications/meaning.
- Different formulations and architectures for GAN have been taken into account for investigation.

Cons:
- There exist some works that propose regularizers to both generator and discriminator, including [Mescheder et al., 2017; Nie & Patel, 2019]. Those regularizers are based on gradient vector fields over the parameters of both discriminator and generator. However, this paper does not provide any discussion about those closely related works and hence places itself in an unclear context.
- Some recent studies [Brock et al., 2019; Zhang et al., 2019] found that regularizing the generator, e.g., using spectral normalization (SN), can help improve training and quality of the generator. SN is a very efficient way to do regularization and can make GAN training much more stable. Therefore, the idea of regularization for generator is not new.
- Since this paper proposes a penalty when training GAN, I expect to see comparisons with some other regularization methods, e.g, SN and gradient penalty. Unfortunately, such a careful comparison is missing. Instead the author(s) used AutoGAN and StyleGAN2, which focus on generator architecture, for investigation. Such experiments do not help much to understand the effectiveness of the proposed regularizer, in comparison with some other approaches. As a result, the significance of this work is unclear.

Minor comments:
- An investigation about the sensitivity of parameter $\lambda$ should be done. It could help us see whether or not the proposed heuristic is effective.
- One motivation of this work is the optimality of the discriminator. However, it is wellknown that the gradient vanishing problem may appear if one train the discriminator to optimality. Therefore the motivation seems not to be supportive.
- Formular (1) seems to be wrong.
- “advAs as an attempt fulfill” in page 4 --> “advAs as an attempt to fulfill”

Reference:
- Andrew Brock, Jeff Donahue, and Karen Simonyan. Large scale gan training for high fidelity natural image synthesis. In International Conference on Learning Representations, 2019.
- Lars Mescheder, Sebastian Nowozin, and Andreas Geiger. The numerics of gans. In Advances in Neural Information Processing Systems, pp. 1825–1835, 2017.
- Weili Nie and Ankit Patel. Towards a better understanding and regularization of gan training dy- namics. In Conference on Uncertainty in Artificial Intelligence (UAI), 2019.
- Han Zhang, Ian Goodfellow, Dimitris Metaxas, and Augustus Odena. Self-attention generative adversarial networks. In International Conference on Machine Learning, pp. 7354–7363, 2019.

---

> ### Author Response · Authors · 2020-11-25
> **Thank you for your thorough review**
>
> - We thank you for referring us to this prior work, and refer to our the general comments above as well as the revised paper for a discussion and our repositioning.
>
> - Regarding "well known that the gradient vanishing problem may appear if one train the discriminator to optimality": This is not true for WGAN and its variations, which we used in this paper. It certainly can be for the original GAN.
>
> - Thank you for pointing out our error in Eq. (1). It has been corrected.

---

### Official Review · AnonReviewer2 · 2020-10-28
**Well-written, but not novel**

**Rating:** 4
**Confidence:** 4

**Review:**

Summary:

The authors observe that many GAN forumations assume optimality of the discriminator at each generator step and clarify this by reviewing a variant of an envelope theorem.
Observing that this optimality condition isn't usually satisfied in practice, the authors propose a generator regularizer term which encourages the discriminator to be approximately optimal by penalizing the norm of the discriminator's gradient vector.
Experiments on MNIST, CIFAR10, and CelebA show that use of the regularizer often yields improvement wrt baselines.

Review:

In summary, the paper's main strength is clarity of exposition; its main weaknesses are that the algorithm isn't novel (see below) and that the experimental results are limited. Overall I can't recommend acceptance in the current form.

Correctness:

The motivation, theory, and algorithms proposed are broadly correct to my knowledge.
I do want to point out that it's possible to obtain an unbiased estimate of $|| \partial_\theta L ||_2^2$ efficiently by computing two independent estimates of the discriminator gradient vector (i.e. using two independent minibatches) and multiplying them. This owes to the fact that $E[X]^2 = E[X]E[X'] = E[X X']$ when $X, X'$ are independent and identically distributed.

Novelty:

The observation that GAN theory sometimes relies on an envelope theorem, and the specific envelope theorem given, aren't particularly novel (for instance, the original GAN paper by Goodfellow made a version of this argument), but they are (in my opinion) under-appreciated in existing literature and so the extra exposition is welcome.

The authors point out that while superficially similar to other gradient penalties, it has a very different motivation, structure, and probably works in a different way; I generally agree with their argument.
However, the proposed regularizer does have exactly the same form (and similar motivation) as the one proposed in https://arxiv.org/abs/1706.04156, so the algorithm isn't novel in the end.

The proposed algorithm is also closely related to https://arxiv.org/abs/1705.10461 .

If this paper is to be accepted, at a minimum it needs a full discussion of the algorithms given in these two works and how they relate to the present work.

Significance:

Empirically the experiments support the usefulness of the proposed algorithm, albeit not to a very great degree.
The baselines compared against aren't state-of-the-art.

One experimental weakness is that in many places the authors claim a reduction in discriminator-steps-per-generator-step as an improvement, but don't separately search over generator/discriminator learning rates in the baseline.
If you want to make this kind of claim, the appropriate baseline is a GAN in which the G and D LRs have been tuned (and tuned independently of each other).
Otherwise, for example, a trivial "penalty" term which just equals the original loss (and hence has the effect of multiplying the generator gradient by two, which is equivalent to multiplying the learning rate by two for SGD) might show improvement.

Clarity:

The paper deserves high marks on clarity in my opinion. It reads easily and contextualizes its work well. The claims made are precise and well-scoped.
Clarity of the exposition is important because it means this paper advances our understanding despite the somewhat limited empirical improvement from the algorithm.

---

> ### Author Response · Authors · 2020-11-25
> **Thank you for your thorough review**
>
> Thank you for appreciating the value our paper brings in terms of exposition. We also thank you for bringing the prior work (both of them) to our attention. As we write for reviewer #2 and in the general comments, we have addressed this in the revised paper and point out that we seek to provide further evidence and bring a new motivation as to why AdvAs is an appropriate regularizer to use.

---

### Official Review · AnonReviewer4 · 2020-10-29
**The proposed regularization was already proposed in a previous work.**

**Rating:** 3
**Confidence:** 5

**Review:**

**Summary of contributions:**
This paper propose a new regularization for training the generator in GANs. They argue that when the discriminator is not optimal this encourages the generator to stay in region where the discriminator is close to optimal. They then propose an heuristic to adaptively control the coefficient of the regularization term. Finally they show empirically that the proposed regularization can indeed improve the generator performance in several setting.

**Major Concern:**
I have a big concern about the contribution of this work, the proposed regularization eq 8 is exactly the regularization also proposed in [[1] Gradient descent GAN optimization is locally stable (NeurIPS 2017)](http://papers.nips.cc/paper/7142-gradient-descent-gan-optimization-is-locally-stable) see eq 4.
This related work is not even mentioned in this paper, also while the perspective is a bit different here, I find the theoretical motivation more rigorous in [1].

**Clarity:**
The paper is quite clear and well written.

**Significance:**
The experiments gives a good overview of the performance of the proposed method, however the results don't always show a major advantage of the proposed regularization for example on CIFAR there doesn't seem to be any benefit. Also this should be compared to other regularization technique for the generator for example Jacobian Clamping proposed in [Is Generator Conditioning Causally Related to GAN Performance? (ICML 2018)](http://proceedings.mlr.press/v80/odena18a.html).
It would also be interesting to have a plot showing the evolution of the adaptative regularization coefficient along training.

**Other comments:**
There is actually a cheap and unbiased estimator for the AdvAs loss by using this property: $||\mathbb{E}[X]||^2 = \mathbb{E}[X_i^TX_j]$ where $X_i$ and $X_j$ are two independent random variables sampled from the same distribution as X. For an example on such an unbiased estimator in a context related to yours see Appendix C of [Stochastic Hamiltonian Gradient Methods for Smooth Games  (ICML 2020)](https://proceedings.icml.cc/static/paper_files/icml/2020/6356-Paper.pdf).

---

> ### Author Response · Authors · 2020-11-25
> **Regarding prior work**
>
> Thank you for pointing us to the referred prior work (please see general comments). We have edited the paper to address this and better position our work. We seek to provide further evidence and bring a new motivation as to why AdvAs is an appropriate regularizer to use.

---

### Official Review · AnonReviewer1 · 2020-11-06

**Rating:** 6
**Confidence:** 2

**Review:**

This paper proposes a new regularizer to improve GAN training. By noticing that the discriminator does not always reach optimum at each iteration, this paper proposes Adversary's Assistant (AdvAs) for helping the discriminator to satisfy this condition. Interestingly, compared to the previous methods for improving GAN training, this work applies the regularizer at the generator (rather than the discriminator) and is theoretical motivated. Experiments on several GAN objectives, datasets and network architectures are provided to support the effectiveness of AdvAs.


*Pros

(1) This paper is clearly written. Even I am not an expert in GAN, I do not encounter too many difficulties in understanding the whole paper.

(2) The whole framework is theoretical motivated. Given that the discriminator is not always at an optimal point during training, this paper derives several theorems and corollaries, which leads to the finding that adding a regularization on the generator could satisfy a necessary condition for training GAN optimally.

(3) Empirical results are provided to show the proposed AdvAs can help GAN training under different settings.



*Cons

(1) This paper uses a minibatch to approximately compute the regulizer $r(\theta,\phi)$. I am wondering if the proposed AdvAs is sensitive to the estimation quality of $r(\theta,\phi)$? For example, if large batch size is used, will the results be better? If yes, then what is the "minimal" batch size to train a good GAN with AdvAs (i.e., outperforms the baseline)?

(2) I appreciate that this paper honestly states that the proposed AdvAs cannot help BEGAN and LSGAN. I encourage the authors to delve deeper into this observed phenomenon and provide a brief discussion on explaining the possible reasons why AdvAs cannot help here.

(3) As the main purpose of AdvAs is to encourage the value of Eq. (7) be close to 0, the authors are encouraged to also plot the value of $D_2h(p_θ, a_φ)$ during the training, as direct evidence for supporting the effectiveness of AdvAs.



**Overall, I think it is an interesting paper, with good theoretical motivation and strong empirical results, therefore I tend to accept it at this time. Nonetheless, I am not an expert in GAN and cannot accurately access the value/importance of this paper. I am open to increase/decrease my score if other expert reviewers provide any positive/negative comments.

---

> ### Author Response · Authors · 2020-11-25
> **Thank you for your helpful comments**
>
> In the response to the first question regarding the batch size: we found that AdvAs was quite robust with respect to the batch size. Without performing any systematic search over the batch size, we did not find any particular sensitivity to it when comparing with and without AdvAs.

---

### Author Response · Authors · 2020-11-25
**General Comments**

We thank you all for your reviews.

Given that AdvAs, as a method, is the same as that found in "Gradient descent
GAN optimization is locally stable" (Nagarajan and Kolter, 2017) and "The
numerics of gans" (Mescheder et al., 2017) we want to address this first and
foremost. We have edited our paper, especially the abstract, introduction and
related work sections, to appropriately refer to this prior work. Unfortunately
this prior work was missed during our background survey, but we would like to
point out that our work is independent of this prior work and that the oversight
was unintentional. We thank the three reviewers who brought this to our
attention and appreciate the thoroughness showcased in this review process.

---

### Decision · Program_Chairs · 2021-01-07
**Final Decision**

**Decision:**

Reject

**Comment:**

The paper received low ratings and the reviewers pointed out a number of issues. The authors' short response failed to address these concerns.